# Biodirected Screening and Preparation of *Larimichthys crocea* Angiotensin-I-Converting Enzyme-Inhibitory Peptides by a Combined In Vitro and In Silico Approach

**DOI:** 10.3390/molecules29051134

**Published:** 2024-03-03

**Authors:** Zhizhi Yang, Changrong Wang, Baote Huang, Yihui Chen, Zhiyu Liu, Hongbin Chen, Jicheng Chen

**Affiliations:** 1College of Food Science, Fujian Agriculture and Forestry University, Fuzhou 350002, China; yzz37741@136.com (Z.Y.); wcr9941@163.com (C.W.);; 2Key Laboratory of Cultivation and High-Value Utilization of Marine Organisms in Fujian Province, Fisheries Research Institute of Fujian, National Research and Development Center for Marine Fish Processing (Xiamen), Xiamen 361013, China; 3Fujian Province Key Laboratory for the Development of Bioactive Material from Marine Algae, Quanzhou Normal University, Quanzhou 362000, China

**Keywords:** *Larimichthys crocea*, ACE-inhibitory peptides, in silico, QSAR, inhibitory kinetics, molecular docking

## Abstract

Food-derived angiotensin-I-converting enzyme (ACE)-inhibitory peptides have gained attention for their potent and safe treatment of hypertensive disorders. However, there are some limitations of conventional methods for preparing ACE-inhibitory peptides. In this study, in silico hydrolysis, the quantitative structure–activity relationship (QSAR) model, LC-MS/MS, inhibition kinetics, and molecular docking were used to investigate the stability, hydrolyzability, in vitro activity, and inhibition mechanism of bioactive peptides during the actual hydrolysis process. Six novel ACE-inhibitory peptides were screened from the *Larimichthys crocea* protein (*LC*P) and had low IC_50_ values (from 0.63 ± 0.09 µM to 10.26 ± 0.21 µM), which were close to the results of the QSAR model. After in vitro gastrointestinal simulated digestion activity of IPYADFK, FYEPFM and NWPWMK were found to remain almost unchanged, whereas LYDHLGK, INEMLDTK, and IHFGTTGK were affected by gastrointestinal digestion. Meanwhile, the inhibition kinetics and molecular docking results were consistent in that ACE-inhibitory peptides of different inhibition forms could effectively bind to the active or non-central active centers of ACE through hydrogen bonding. Our proposed method has better reproducibility, accuracy, and higher directivity than previous methods. This study can provide new approaches for the deep processing, identification, and preparation of *Larimichthys crocea*.

## 1. Introduction

Hypertension is a multifactorial chronic non-communicable disease, the prevalence of which is directly related to our daily diet, and has become one of the most discussed public health problems in recent decades [1]. The development of hypertension greatly increases the risk of various arterial diseases, including atherosclerosis, myocardial infarction, coronary heart disease, stroke, and renal failure [2]. In the renin–angiotensin system, ACE is a key enzyme that catalyzes the transformation of potent pressor angiotensin II (Ang II) from non-pressor angiotensin I (Ang I), which raises blood pressure [3]. Therefore, effective inhibition of ACE activity is considered one of the treatments for hypertensive disorders. However, ACE inhibitors are generally chemically synthesized, and long-term use of chemically synthesized ACE inhibitors is prone to side effects such as renal impairment, rash, and angioneurotic edema [4]. With the growing awareness of people’s self-health, ACE-inhibitory peptide, which is safe, free of side effects, and purely natural, is increasingly favored by patients and medical professionals [5].

Bioactive peptides are particular protein fragments made up of two to twenty amino acids that can be extracted or predicted from plant and animal proteins using fermentation, enzymolysis, or bioinformatics [6]. The field of bioinformatics, sometimes referred to as in silico analysis, is created when computational science, applied mathematics, and biology come together [7]. The methods and tools based on in silico analysis have been successfully applied to the development of bioactive peptides in recent years [5]. For example, Ngoh et al. [8] successfully identified peptides with high ACE-inhibitory activity from the pinto bean using bioinformatics tools such as BIOPEP-UWM, PeptideRanker, and AutoDock. Li et al. [9] used bioinformatics tools such as Peptide Ranker, AHTPDB, and ADME to isolate two highly active ACE-inhibitory peptides from the edible symbiotic *Boletus griseus*–*Hypomyces chrysospermus*. It is undeniable that the advent of bioinformatics has pushed the development of bioactive peptides. However, some of the in silico hydrolysis-released peptides could not be replicated by in vitro experiments due to the complexity of enzyme-protein interactions during actual hydrolysis. In addition, many studies lacked validation of the activity of bioactive peptides obtained by in silico hydrolysis screening, resulting in low confidence in the bioactive peptides they prepared. Therefore, this study combined bioinformatics and in vitro experiments to jointly search for effective ACE-inhibitory peptides, considering the limitations of traditional extraction methods and in silico hydrolysis. This improved the accuracy and reliability of in silico prediction and provided a theoretical and experimental basis for high-throughput screening of bioactive peptides in protein hydrolysates.

Research has shown that protein-rich marine animals have proven to be an important source of potentially bioactive peptides [10]. For example, Lee et al. [11] obtained an ACE-inhibitory peptide from tuna frame using enzymatic digestion, ion exchange chromatography, HPLC, and Q-TOF mass spectrometry, and the peptide significantly reduced systolic blood pressure in spontaneously hypertensive rats after oral administration. Fu et al. [12] identified and screened three umami peptides from the Pacific oyster using nano-HPLC-MS/MS, iUmami-SCM, and molecular docking methods. Tsai et al. [13] found that the protein hydrolysate of hard clam had high ACE-inhibitory activity by hydrolysis extraction, size-exclusion chromatography, and sequence analysis. In conclusion, marine-derived proteins such as fish, hard clams, oysters, squid, tuna, sea urchins, and shrimp are important sources of bioactive peptides [14]. *Larimichthys crocea*, also called large yellow *croaker* or *croceine croaker*, is a member of the *Sciaenidae* family and the order Perciformes and is primarily found in the Northwest Pacific region [15]. Because of its delicious and nutritious flesh, *Larimichthys crocea* is regarded as one of the marine fish with the highest commercial value in the marine fishing business. As a result, aquaculture production in China, Japan, and other Asian nations has been rising annually [16]. *Larimichthys crocea* is rich in a variety of trace elements, unsaturated fatty acids, 17 kinds of amino acids, proteins, and other nutrients [17]. Especially the protein, which contains up to 18%, and is thus considered a high-quality protein source [18]. The bioactive peptides extracted and purified from the muscle of *Larimichthys crocea* have antioxidant [19], antibacterial [20], anticancer [21], and other bioactive functions. Nevertheless, there has not been any additional research conducted on the isolation of ACE-inhibitory peptides from *Larimichthys crocea* protein (*LC*P). Thus, the production of ACE-inhibitory peptides from *LC*P in this work serves as a reference for broadening the protein origins of ACE-inhibitory peptides in addition to aiding in the thorough processing of *Larimichthys crocea*.

In summary, the objectives of this investigation are as follows: (1) Construction of a screening system for ACE-inhibitory peptides based on biological information such as gene sequences and protein sequences corresponding to *LC*P, combined with QSAR modeling and in vitro enzymatic digestion. (2) Synthesize peptides with high ACE-inhibitory activity and analyze their stability through simulated in vitro digestion. (3) Elucidation of the binding mode between ACE-inhibitory peptides and ACE using inhibition kinetics and molecular docking. This study provides a theoretical basis for screening bioactive peptides with high activity from animal proteins.

## 2. Results and Discussion

### 2.1. Analysis of ACE-Inhibitory Activity of Different Molecular Weight Fractions in Larimichthys crocea Protein Enzymatic Hydrolysates

The inhibitory activity of ACE-inhibitory peptides in hydrolysates has been reported to be related to their molecular weights [22]. The inhibitory effects of the *LC*P digest and different molecular weight fractions on ACE are shown in Figure 1. The other peptide fractions with lower molecular weights showed lower IC_50_ values and higher rates of ACE inhibition compared to the <5 kDa component. This could be explained by the fact that smaller molecular weight peptides were able to insert into the clefts of the active site and bind with catalytically relevant amino acids, interfering with the ACE catalytic substrate, while larger molecular weight peptides were unable to enter the spatially narrow active centers to bind with amino acids in the ACE active site [23]. There are some differences in ACE-inhibitory activity between different molecular weights. The ACE-inhibitory activity of molecular weight fractions <3 kDa was higher than that of other molecular weights, and the ACE inhibition rate and IC_50_ were 96.29% and 0.26 mg/mL, respectively, suggesting that the fractions with the strongest ACE-inhibitory activity in the products of the *LC*P digest were concentrated in the molecular weights of less than 3 kDa; this is consistent with previous findings that most functional peptides with ACE-inhibitory activity are concentrated in small peptides with molecular weights below 3 kDa [24]. Therefore, the group of <3 kDa was selected for LC-MS/MS analysis in this study.

### 2.2. QSAR Prediction Model

QSAR modeling is a bioinformatics method that uses mathematical models to describe the relationship between the structure of a molecule and a certain biological activity of the molecule, and this relationship can be utilized to further predict the value of a specific biological activity [5]. Partial least-squares regression (PLS) was used to analyze the association between log-transformed IC_50_ values (Y) and amino acid descriptors (predictors, X), leading to the establishment of a two-component PLS model. To obtain a high correlation coefficient (R^2^), we exclude outliers on the basis of the t,u-score plots of the PLS model and finally construct the relationship plots between predicted and measured values (Figure 2). In particular, the multiple correlation coefficient R^2^ of the tripeptide model was 0.6478, which is a significant improvement in the quality of the model compared to the value of the tripeptide model established in the previous study (R^2^ = 0.4) [25]. Meanwhile, our models of hexapeptide (R^2^ = 0.8355), heptapeptide (R^2^ = 0.7623), and octapeptide (R^2^ = 0.8542) also outperformed the models of hexapeptide (R^2^ = 0.6810), heptapeptide (R^2^ = 0.7120), and octapeptide (R^2^ = 0.7930) established by Wu et al. [26]. This may be because some newer data were chosen for the modeling process in this experiment or because the peptide IC_50_ values used for modeling were smaller. Therefore, all four models developed in this experiment have good predictive ability and can be used to predict the activity values of ACE-inhibitory peptides obtained from bioinformatics-based in vitro assay screening.

### 2.3. Identification of Potential ACE-Inhibitory Peptides

Hydrolysis of proteins is a usual method to gain bioactive peptides [27]. To accurately tap into effective ACE-inhibitory peptides, we investigated the release of effective ACE-inhibitory peptides from *LC*P catalyzed by papain–trypsin by in silico hydrolysis and in vitro hydrolysis. The results are shown in Figure 3, in which 21 peptides (3 tripeptides, 5 hexapeptides, 4 heptapeptides, and 9 octapeptides) from the set of peptides released by in silico hydrolysis were present in the hydrolysate with a molecular weight <3 kDa. Among them, we found that YNL (Tyr-Asn-Leu) peptides have been reported to have ACE-inhibitory activity, antioxidant activity, and renin-inhibitory activity [28]. In addition, the peptides obtained by in silico hydrolysis exhibit similarity with the peptides obtained by actual hydrolysis, suggesting that some of the predicted peptides can indeed be obtained by actual trypsin–papain cohydrolysis. The in silico hydrolysis release of *LC*P was the result of in silico hydrolysis, whereas the peptides released by in silico hydrolysis could be reproduced experimentally, demonstrating that these peptides have good stability. Therefore, in view of the complexity of the in vitro hydrolysis of proteins, this study utilizes in silico hydrolysis and in vitro experiments to jointly search for potential ACE-inhibitory peptides, which can achieve an accurate match between predicted peptides and actual generated peptides and is conducive to bridging the gap between in silico and in vitro experiments.

### 2.4. Prediction and Screening of ACE-Inhibitory Peptide Activity

The 21 peptides obtained by in silico hydrolysis and in vitro experiments were input into the corresponding QSAR models, and the IC_50_ values of each peptide after prediction are shown in Table 1. The IC_50_ values of all twenty-one peptides were low; sixteen of them had projected values less than 100 μM, and six of them had values less than 10 μM. This not only indicates that *LC*Ps are good precursors of ACE-inhibitory peptides but also demonstrates that validation of in silico coupled experiments can be an effective method for ACE-inhibitory peptide discovery. Within the QSAR-predicted peptides, YNL has been determined to have ACE-inhibitory activity, which has not been reported for the other peptides generated in this investigation. Therefore, based on the inverse relationship between the activity of ACE-inhibitory peptides and IC_50_ values [29], six peptides with IC_50_ values less than 10 μM were selected for further study. Standards of these six peptides were synthesized by analysis and injected according to the liquid-phase conditions and mass spectrometry parameters. The peptide sequences were identified using LC-MS/MS. By matching the b- and y-series ions with those recorded in the database, the sequences of the six representative peptides were obtained as Ile-Pro-Tyr-Ala-Asp-Phe-Lys (IPYADFK), Leu-Tyr-Asp-His-Leu-Gly-Lys (LYDHLGK), Ile-Asn-Glu-Met-Leu-Asp-Try-Lys (INEMLDTK), Ile-His-Phe-Gly-Thr-Thr-Gly-Lys (IHFGTTGK), Asn-Trp-Pro-Trp-Met-Lys (NWPWMK), and Phe-Tyr-Glu-Pro-Phe-Met (FYEPFM). Furthermore, we have checked the purity of each synthesized peptide to be over 98% by HPLC, which is up to our experimental requirements.

### 2.5. Validation of ACE-Inhibitory Peptide Activity In Vitro

It has been demonstrated that the type and sequence of amino acids affect how strongly peptides bind to ACE [30]. For instance, ACE-inhibitory peptides bind more strongly to ACE when Pro (P), Tyr (Y), Lys (K), Phe (F), Leu (L), Val (V), Gly (G), Tyr (Y), Ala (A), His (H), Phe (F), Ile (I), and Grn (Q) are present, or more strongly to ACE when Pro (P), Tyr (Y), Lys (K), Phe (F), Leu (L), Val (V), Ile (I), Ala (A), and Gly (g) are present. [3]. The six peptides with the lowest activity predicted by the QSAR model were assayed for activity, and the results are shown in Table 2. The IC_50_ values of the six peptides were 0.63 ± 0.04 μM to 10.26 ± 0.21 μM, all of which indicated high ACE-inhibitory activity. Among them, the ACE-inhibitory activity of IPYADFK was higher than that of the other five peptides, which can indicate the important contribution of Ile (I), Pro (P), Tyr (Y), Ala (A), and Phe (F) to the ACE-inhibitory peptides. Currently, a few novel antihypertensive peptides with ACE-inhibitory activity have been separated and obtained from protein hydrolysates of marine fish animal origin. For example, VVLASLK and LEPWR (IC_50_ values of 961.5 μM and 99.5 μM) from Pacific saury [31], and SP, VDRYF, VHGVV, YE, FEM, and FWRV from protein hydrolysate of skipjack tuna muscle (IC_50_ values of 0.06 ± 0.01 mg/m to 2.18 ± 0.20 mg/mL) have been reported to have ACE-inhibitory activity [32]. The six ACE-inhibitory peptides identified in this study had lower IC_50_ values compared to these ACE-inhibitory peptides, indicating that the six peptides had high ACE-inhibitory activity and were close to the predicted data of the QSAR mode. In summary, the QSAR model has good predictive ability, and our method has high feasibility.

### 2.6. Stability Analysis of ACE-Inhibitory Peptides

Bioactive peptides must resist degradation by gastrointestinal enzymes to reach target organs and tissues and exert their functional properties in vivo [33]. Therefore, changes in peptide activity after gastrointestinal digestion of six ACE-inhibitory peptides obtained on the basis of QSAR modeling were determined to analyze their antidigestive properties. As shown in Table 2, the activities of the digestion products of IPYADFK, FYEPFM, and NWPWMK did not change significantly from those before digestion (*p* < 0.05). This is in accordance with earlier research showing that peptides’ antidigestive qualities are related to their structural makeup and that proline-containing peptides are more resistant to digestion by the gastrointestinal tract [34]. The activities of LYDHLGK, INEMLDTK, and IHFGTTGK were significantly reduced (*p* < 0.05), but their digestion products still showed significant ACE-inhibitory activity. It is speculated that during digestion, LYDHLGK, INEMLDTK, and IHFGTTGK may be broken down into smaller peptide fragments. The increase in the IC_50_ values of LYDHLGK, INEMLDTK, and IHFGTTGK after digestion may be caused by synergistic or antagonistic interactions amongst the peptides in the digested combination [35]. To summarize, during the simulated intestinal digestion, the ACE-inhibitory activity of the IPYADFK, FYEPFM, and NWPWMK peptides remained constant, while the ACE-inhibitory peptides of the LYDHLGK, INEMLDTK, and IHFGTTGK are degraded by the digestive system.

### 2.7. Inhibition Kinetics Study

Studying the nature of the inhibition kinetics of ACE-inhibitory peptides is one of the main approaches to elucidate their mechanism of action [36]. There are several forms of inhibition between ACE-inhibitory peptides and ACE, including competitive inhibition, non-competitive inhibition, anticompetitive inhibition, and mixed inhibition, as a result of the inhibitory peptides’ distinct structures [37]. Figure 4 displays the findings of this experiment, which used Lineweaver–Burk plots to examine the inhibitory patterns of the six peptides. IPYADFK, LYDHLGK, INEMLDTK, IHFGTTGK, and NWPWMK can be judged as competitive ACE inhibitors on the basis of the enzyme inhibition kinetic parameters of constant V_max_ and increasing K_m_; this indicated that IPYADFK, LYDHLGK, INEMLDTK, IHFGTTGK, and NWPWMK can bind directly to the active site of ACE or form peptide-ACE complexes for the purpose of inhibiting ACE activity [38]. FYEPFM was determined to be a non-competitive ACE inhibitor based on the inhibition curves of two different concentrations of FYEPFM intersected with the control on the X-axis. This shows that FYEPFM may combine with the inactive site of ACE and reduce the catalytic activity of ACE. This is consistent with previous studies, where FDGSPVGY can competitively combine with the center site of ACE, whereas VFDGVLRPGQ combines with other sites of ACE and exhibits non-competitive inhibition [39]. In addition, the ACE-inhibitory peptides from marine fish sources all exhibited different forms of inhibition. For example, LEPWR obtained from Pacific saury showed a mixed competitive inhibition [31], and VGLFPSRSF obtained from tilapia skin showed non-competitive inhibition [40]. Therefore, these six peptides can be further investigated and validated for their inhibitory mechanisms by molecular docking.

### 2.8. Molecular Interactions between Peptides and ACE

Molecular docking is primarily a computerized method for studying ligand–receptor interactions [41]. Only the inhibition of ACE by peptides was investigated through inhibition kinetics, failing to show a specific mechanism of inhibition between the peptides and ACE. Consequently, the molecular docking between the peptides and ACE was examined using AutoDock 4.2. The six peptides that were docked with ACE had binding energies of −10.5 kJ/mol, −9.7 kJ/mol, −8.8 kJ/mol, −9.9 kJ/mol, −10.3 kJ/mol, and −9.6 kJ/mol, respectively. This suggests that all six peptides were able to bind to ACE with effectiveness. The three main active binding pockets of ACE are S_1_ (Ala354, Glu384, and Tyr523 residues), S_2_ (Gln281, His353, Lys511, His513, and Tyr520 residues) and S′ (Glu162) [35]. Figure 5 shows the optimal molecular docking conformations of the six peptides and ACE, with IPYADFK forming the most (14) hydrogen bonding forces with Glu384, Tyr523, Ala354, Gln281, His353, Lys511, Try520 Tyr360, Glu403, Arg402, Ala356, His387, Asn66, and the metal receptor (Zn701). This is consistent with the lowest IC_50_ value and docking binding energy, indicating that the molecular docking results are consistent with in vitro experiments. Notably, LYDQHLGK, INEMLDTK, IHFGTTGK, and NWPWMK all bind to amino acid residues in the S_1_, S_2_, and S′ pockets of ACE via hydrogen bonding, which suggests that IPYADFK, LYDQHLGK, INEMLDTK, IHFGTTGK, and NWPWMK inhibit ACE activity by embedding themselves in the catalytic cavity of ACE and competing with the substrate. Instead of interacting with key amino acid residues in the active center site of ACE, FYEPFM exhibited high ACE-inhibitory activity by binding to Zn^2+^ of ACE through the thiol group of histidine; this is consistent with the findings of Shao et al. [42] that the non-competitive peptide DIGGL cannot bind to Zn701 and the active pockets of ACE via conventional hydrogen bonding. Meanwhile, the above findings are consistent with the determination of its ACE-inhibition kinetic pattern.

The two-dimensional structure diagrams in Figure 5 show that more hydrogen bonds, hydrophobic interactions, and electrostatic interactions were formed between the amino acid residues of ACE and the six peptides. This suggests that hydrogen bonding, hydrophobic interactions, and electrostatic interactions are the main drivers of the binding of IPYADFK, LYDQHLGK, INEMLDTK, IHFGTTGK, NWPWMK, and FYEPFM to ACE. This result is consistent with the conclusion of Ma et al. [43] that hydrogen bonding, electrostatic interactions, and hydrophobic interactions are the main forces stabilizing ACE-LPGPGP complexes.

## 3. Materials and Methods

### 3.1. Materials

*Larimichthys crocea* was obtained from a supermarket in the area. The muscles of the back were collected, chopped using a meat grinder (SUPOR Co., Ltd., Zhejiang, China), and stored at –20 °C until use. Hippuryl-histidyl-leucine (HHL), hippuric acid (HA), and ACE were obtained from Sigma-Aldrich (Milwaukee, WI, USA). Pepsin (400 U/mg), papain (8.0 × 10^5^ U/g), and trypsin (2.5 × 10^6^ U/g) were purchased from Macklin Biotechnology Co., Ltd. (Shanghai, China). Reagents for HPLC and LC-MS/MS analysis as HPLC grade. All other chemicals employed are of chemical analytical grade.

### 3.2. In Silico Hydrolysis

*Larimichthys crocea*’s representative protein myosin heavy chain (A0A6G0IXB6) was acquired from the UniProt database (https://www.uniprot.org/, accessed on 7 January 2023). Trypsin and papain were selected with the “Enzyme action” tool of the BIOPEP-UWM database (http://www.uwm.edu.pl/biochemia/index.php/pl/biopep, accessed on 9 January 2023) for combined hydrolysis, and the peptides obtained by hydrolysis were sorted and classified.

### 3.3. Preparation of Larimichthys crocea Protein Extract

A low concentration of phosphate buffer solution was initially added to muscles of the back in order to remove water-soluble proteins; then, myofibrillar proteins were extracted using a high concentration of phosphate buffer solution [44]. According to the previous method with slight modification [45], optimized papain (600 U/g) and trypsin (600 U/g) were selected for coenzymatic digestion (enzyme reaction conditions: The ratio of enzyme to crude extract protein content was 1:6.3, pH = 8). After 6 h of digestion on a constant temperature shaker (HZ-C, Boyi Xunshi Co., Shanghai, China) at 45 °C, the supernatant was taken and placed in a centrifuge equipped (Avanti J-26XP, ShanghaiAnting Electronic Apparatus Factory, Shanghai, China) with a No. 9 rotor (12 × 50 mL) and centrifuged at 8000× *g* and 4 °C for 20 min. An ultrafiltration device (Vivaflow 200 Minimate, Sartorius, Gottingen, Germany) was used to separate seven different molecular weight components: <0.5 kDa (the duration of the flow was 32 h), 0.5 kDa–1 kDa (the duration of the flow was 30 h), <1 kDa (the duration of the flow was 28 h), 1 kDa–3 kDa (the duration of the flow was 25 h), <3 kDa (the duration of the flow was 24 h), 3 kDa–5 kDa (the duration of the flow was 22 h), and <5 kDa (the duration of the flow was 12 h), the rotation speed of the constant flow pump was 60 rpm, and the enzymatic solution was 300 mL. The samples were lyophilized in a freeze-dryer (LGJ-1D-80, Shanghai Sheyan Instrument Co. Shanghai, China) for 24 h and then placed in a refrigerator at –80 °C.

### 3.4. Determination of ACE-Inhibitory Activity

The methods were previously established in our laboratory [46]. The system had a total volume of 120 μL. Then, 60 μL of peptide samples (1 mg/mL) and 20 μL of ACE solution (0.1 U/mL) were placed in a 37 °C water bath for 10 min, and 50 μL of hippuryl-histidyl-leucine (HHL) solution (1 mM) was added to react for 60 min. Finally, the reaction was terminated by adding 1 M HCL (100 μL). The sample solution of the determination group was replaced by 60 μL boric acid buffer solution (pH = 8.3 0.3 M NaCl) in the control tube. Then, 10 μL boric acid buffer was put into the blank control tube instead of the ACE solution used in the control group. Column chromatography conditions were as follows: Waters-C18, 5 μm, 4.6 mm × 250 mm; detection wavelength: 228 nm; mobile phase: 80% A: 0.1% formic acid + 0.1% trifluoroacetic acid + 98% deionized water, 20% B: 100% acetonitrile; flow rate: 0.8 mL/min; column temperature: 30 °C; injection volume: 10 μL. The ACE-inhibition rate was calculated by Formula (1):(1)$$ACEinhibitoryrate\left(\%\right)=\frac{\left(\left[HA\right]c-\left[HA\right]s\right)}{\left([HA]c-[HA]H\right)}\times 100\%$$
where *HA* is the concentration of hippuric acid, *[HA]c* is the concentration of *HA* in the control sample, *[HA]s* is the sample *HA* concentration, and *[HA]H* is the blank control *HA* concentration. The IC_50_ value was defined as the inhibitor concentration inhibiting 50% of ACE activity. First, the in vitro ACE inhibition rate was determined for samples of different concentrations. Then, the fitting curve and equation were drawn with the sample mass concentration as abscissa and ACE inhibition rate as ordinate. The value of IC_50_ was calculated according to the fitted curve.

### 3.5. QSAR Modeling to Evaluate Potential ACE-Inhibitory Peptides in Larimichthys crocea Protein

Constructing QSAR models with reference to previous studies [26]. Four ACE-inhibitory peptide datasets containing 260 tripeptides, 101 hexapeptides, 78 heptapeptides, and 65 octapeptides were constructed. The data of these peptides were obtained from previous studies [47] and the DFBP database (http://www.cqudfbp.net/, accessed on 15 March 2023). The IC_50_ value of the peptides selected for modeling needs to be below 15 mM. The structural characteristics of the peptides involved were characterized by the amino acid descriptor symbol 5z-scale (Table 3). Amino acid property parameters were collected from AAindex, a database of physicochemical properties of amino acids, and principal component analysis was performed to obtain five z-scores (z1, z2, z3, z4, and z5). The lipophilic qualities are represented by the z1 descriptor, the spatial arrangement properties of amino acids by z2, the charged properties of amino acids by z3, and other properties like electronegativity, heat of formation, electrophilicity, and hardness by z4 and z5. The peptides were predicted by partial least-squares (PLS) regression using SIMCA-P14.1 software.

### 3.6. Nano LC-MS/MS Analysis

Hydrolysates were re-dissolved in solvent A (A: 0.1% formic acid aqueous solution) and analyzed with a Q-Exactive Plus and EASY-nanoLC 1200 system (Thermo Fisher Scientific, Waltham, MA, USA). The sample was loaded onto a 25 cm analytical column (75 μm inner diameter, 1.9 μm resin (Dr Maisch)). The column flow rate was maintained at 300 nL/min with a column temperature of 40 °C. The electrospray voltage was set to 2 kV. Mobile phases used for online isolation consisted of 0.1% formic acid in water and 0.1% formic acid with acetonitrile solution. The data-dependent acquisition mode was used on the mass spectrometer, which automatically alternated between MS and MS/MS modes. Using the Orbitrap, 70,000 resolution full scan MS spectra (*m*/*z* 200–2000) were surveyed. The maximum injection time was 50 ms, and the automated gain control target was 3 × 10^6^. The largest time of injection for HCD-MS/MS was 45 ms, the dynamic exclusion was 30 s, the automatic gain control target was 1 × 10^5^, and the MS/MS resolution was placed at 17,500. The database was *Larimichthys crocea* (version 2023, 23,962 entries), which was downloaded from the UniProt database (https://www.uniprot.org/, accessed on 3 May 2023).

### 3.7. Synthesis of Peptides

The identified peptides were synthesized using solid-phase peptide synthesis methods by Jiangsu Jinsirui Biotechnology Co., Ltd. (Nanjing, China). The purity of the obtained polypeptide was >98%, as determined by high-performance liquid chromatography, and the sequence was verified by LC-MS/MS analysis.

### 3.8. The Stabilization of ACE-Inhibitory Peptides during Simulated Digestion In Vitro

A slight modification of the research methodology based on the previous study [48] was implemented. With slight modification, the peptide samples were dissolved in deionized water (1 mg/mL), adjusted the pH to 2 with 1 M HCl, pepsin (2% *w*/*w*) was added, and the reaction was run for 2 h at 37 °C in the temperature shaker. The reaction was adjusted to pH = 7.5 with 2 M NaOH, and trypsin (2% *w*/*w*) was added for 2 h. Then, it was placed in boiling water at 100 °C for 10 min and finally removed to wait for the temperature to decrease to room temperature and centrifuged to detect their ACE activity.

### 3.9. Inhibitory Kinetics Study

Minor modifications with reference to previous research methodology [29] and different substrate concentrations of HHL (0.5 mM, 1 mM, 2 mM, 4 mM, and 5 mM) were used to react with different concentrations of peptides (0 mg/mL, 0.2 mg/mL, and 0.5 mg/mL). A double-inverse Lineweaver–Burk plot was made using the inverse of the rate of HA production as the y-axis and the inverse of the concentration of the substrate HHL as the x-axis.

### 3.10. Molecular Docking

Some modifications were made based on previous research methods [23]; the ACE crystal structure (PDB ID: 1O86) was obtained from the RCSB Protein Data Bank (http://www.rcsb.org/, accessed on 20 May 2023). Discovery Studio 4.5 and AutoDock Tools 1.5.6 were used to process the protein structures, and the processed structures were used as initial models for molecular docking. The molecular docking software AutoDock4.2 was used to calculate the semiflexible molecular docking of ACE and active skin. The Lamarck genetic algorithm was used to optimize the molecular docking energy, and 100 docking simulations were carried out. The interaction between ACE-inhibitory peptides and ACE protein molecules (hydrogen bonds, hydrophilic/hydrophobic forces, etc.) was analyzed by visualization and analysis software such as AutoDock Tools 1.5.7, PyMOL 2.6, and Discovery Studio 4.5.

### 3.11. Statistical Analysis

All experiments were repeated in triplicates, the results were displayed as mean ± standard deviation, and the data were analyzed statistically using SPSS Statistics 22 software (SPSS Inc., Chicago, IL, USA) (*p* < 0.05).

## 4. Conclusions

In summary, this study is based on a combination of bioinformatics and in vitro experimental verification to target, screen, and identify antihypertensive peptides. The proposed method has been successfully applied to the screen of ACE-inhibitory peptides derived from *Larimichthys crocea*, which provides a good start for the systematic and high-throughput identification of bioactive peptides. We also investigated ACE-inhibitory peptides screened based on this method in terms of inhibition forms and the mode of binding. PYADFK, LYDHLGK, INEMLDTK, IHFGTTGK, and NWPWMK, which are competitive inhibitors, can effectively bind to the active center of ACE through hydrogen bonds and have low IC_50_ values. The non-competitive inhibitor (FYEPFM) binds to the non-central active site of ACE and reduces the catalytic activity of ACE. The study of the inhibition mechanism of ACE-inhibitory peptides further confirmed that the bioinformatics coupling experiment is an effective method for high-throughput screening of antihypertensive peptides and lays a theoretical foundation for the development of potential antihypertensive functional food components. However, future studies are needed to determine the in vivo activity of these novel ACE-inhibitory peptides to assess the potential of these predicted peptides as alternative food ingredients.

## Figures and Tables

**Figure 1 molecules-29-01134-f001:**
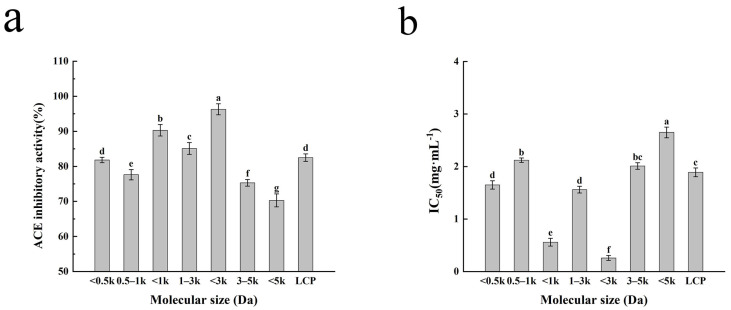
The ACE-inhibitory activity of the fractions with different molecular weights and *Larimichthys crocea* protein hydrolysate (**a**). The IC_50_ values of different molecular weights and *Larimichthys crocea* protein hydrolysate (**b**). Different letters on the same figure indicate significant differences (*p* > 0.05). The values represent the average of *n* = 3, and the standard deviation of each test result is represented by an error bar.

**Figure 2 molecules-29-01134-f002:**
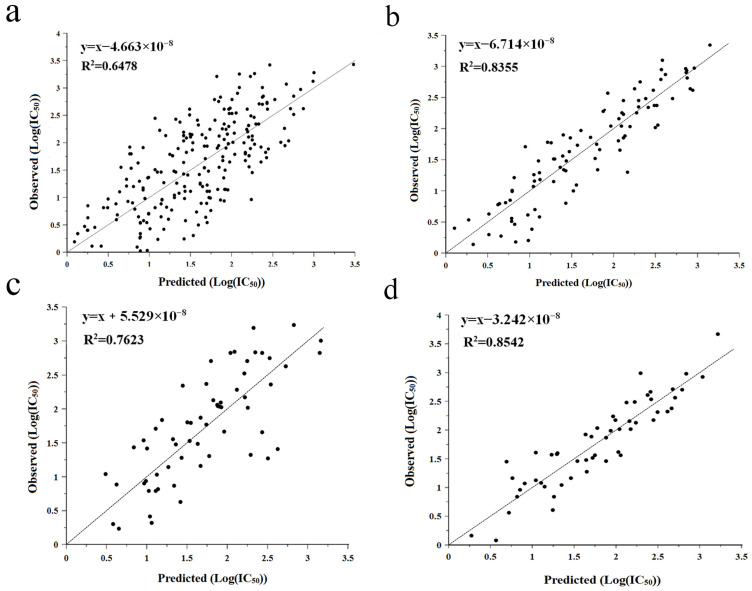
QSAR modeling was performed to investigate the relationship between observed and predicted values of active peptides, including tripeptide (**a**), hexapeptide (**b**), heptapeptide (**c**), and octapeptide (**d**).

**Figure 3 molecules-29-01134-f003:**
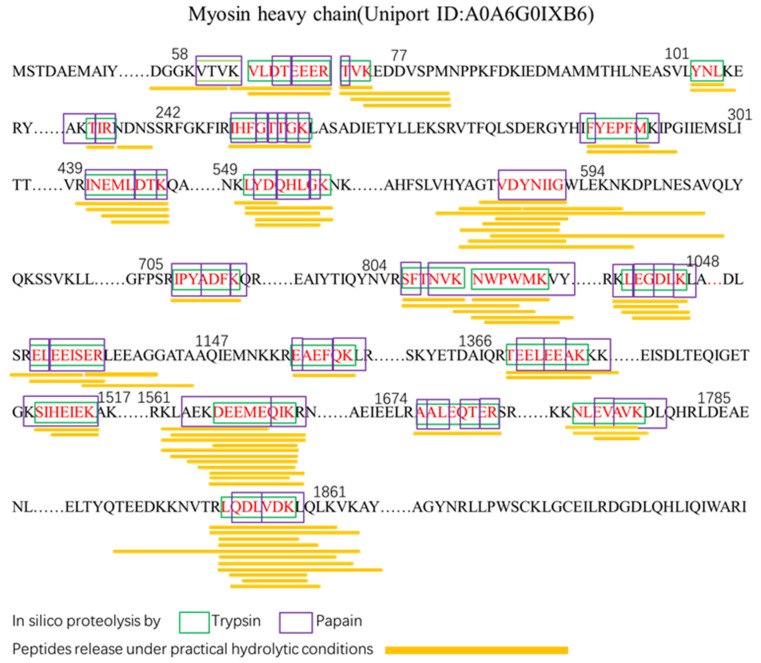
Effective bioactive peptides released by in silico hydrolysis and in vitro protein digestions of *Larimichthys crocea* protein.

**Figure 4 molecules-29-01134-f004:**
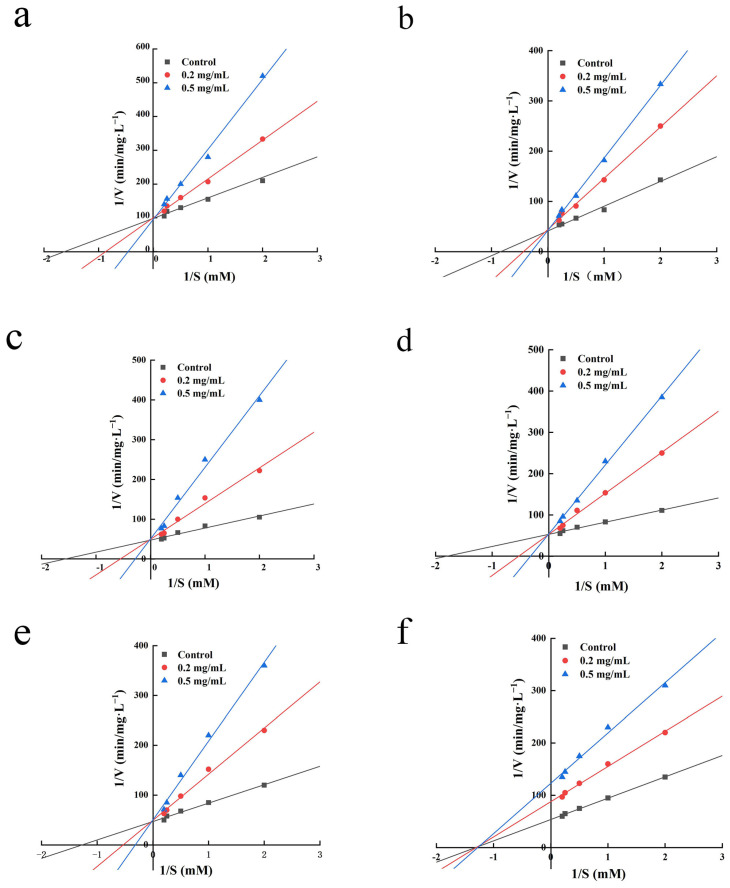
Lineweaver–Burk plots of ACE inhibition for IPYADFK (**a**), LYDHLGK (**b**), INEMLDTK (**c**), IHFGTTGK (**d**), NWPWMK (**e**), and FYEPFM (**f**).

**Figure 5 molecules-29-01134-f005:**
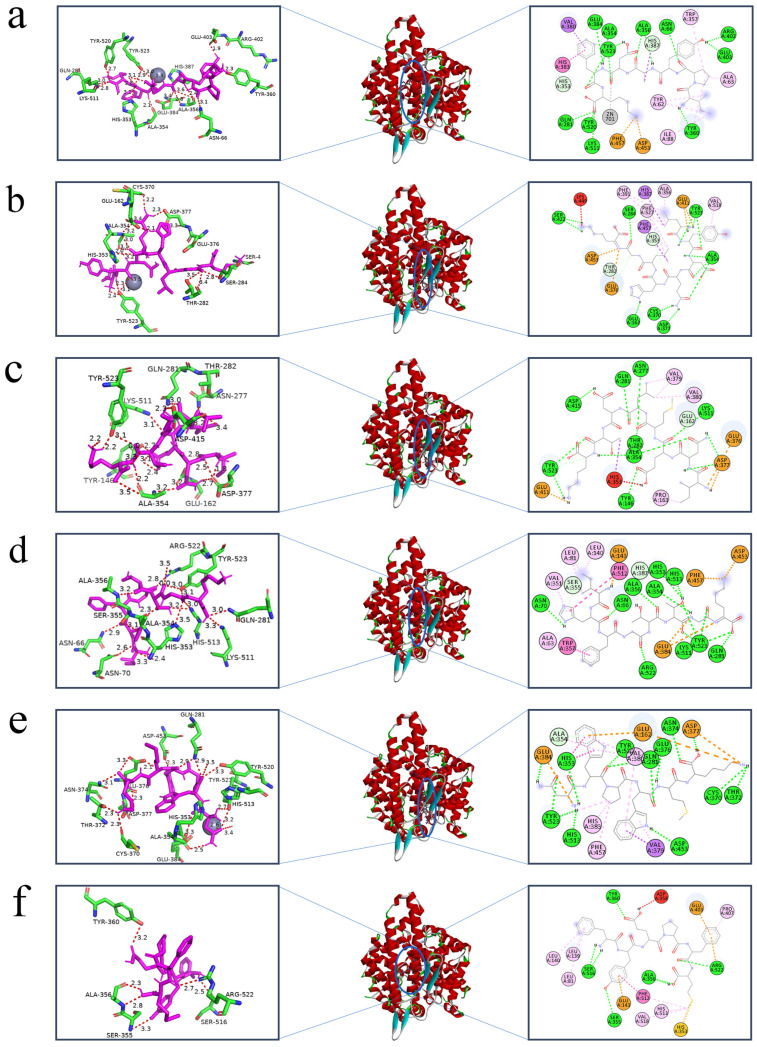
The 3D structures, overviews, and 2D representations of the best docking sites for peptides in molecular docking: IPYADFK (**a**), LYDHLGK (**b**), INEMLDTK (**c**), IHFGTTGK (**d**), NWPWMK (**e**), and FYEPFM (**f**). Peptides are labeled in purple, amino acid residues at the ACE docking site are labeled in green, and hydrogen bonds are labeled in red (where the numbers indicate the length of the hydrogen bonding force). The green dotted lines in the 2D diagrams are hydrogen bonds.

**Table 1 molecules-29-01134-t001:** Predicted IC_50_ values by QSAR model.

Peptides	IC_50_ (µM)	Peptides	IC_50_ (µM)	Peptides	IC_50_ (µM)
TVK	205.63	NWPWMK	8.19	AALEQTER	37.25
TIR	219.84	IPYADFK	0.64	INEMLDTK	1.82
YNL	43.40	SIHEIEK	23.51	DEEMEQIK	30.98
EAEFQK	416.43	LQDLVDK	112.97	NLEVAVK	11.29
FYEPFM	9.53	VDYNIIG	15.26	VLDTEEER	16.77
SFTNVK	52.93	TEELEEAK	12.74	IHFGTTGK	7.87
LEGDLK	233.12	LYDQHLGK	3.25	ELEEISER	56.22

**Table 2 molecules-29-01134-t002:** ACE-inhibitory activity of synthetic peptides before and after in vitro digestion simulation.

Peptides	IPYADFK	LYDHLGK	INEMLDTK	IHFGTTGK	NWPWMK	FYEPFM
IC_50_ value before digestion (µM)	0.63 ± 0.09 ^a^	9.41 ± 0.24 ^b^	6.53 ± 0.21 ^b^	5.34 ± 0.15 ^b^	6.23 ± 0.34 ^a^	10.26 ± 0.21 ^a^
IC_50_ value after digestion (µM)	0.80 ± 0.08 ^a^	24.34 ± 0.29 ^a^	15.65 ± 0.15 ^a^	33.73 ± 0.23 ^a^	7.27 ± 0.17 ^a^	11.41 ± 0.43 ^a^

The distinct alphabets within the columns indicate significant differences at *p* < 0.05.

**Table 3 molecules-29-01134-t003:** Descriptors for characterizing amino acids.

Abbrev.	Name	z1	z2	z3	z4	z5
Ala	A	0.24	−2.32	0.60	−0.14	1.30
Arg	R	3.52	2.50	−3.50	1.99	−0.17
Asn	N	3.05	1.62	1.04	−1.15	1.61
Asp	D	3.98	0.93	1.93	−2.46	0.75
Cys	C	0.84	−1.67	3.71	0.18	−2.65
Gln	Q	1.75	0.50	−1.44	−1.34	0.66
Glu	E	3.11	0.26	−0.11	−3.04	−0.25
Gly	G	2.05	−4.06	0.36	−0.82	−0.38
His	H	2.47	1.95	0.26	3.90	0.09
Ile	I	−3.89	−1.73	−1.71	−0.84	0.26
Leu	L	−4.28	−1.30	−1.49	−0.72	0.84
Lys	K	2.29	0.89	−2.49	1.49	0.31
Met	M	−2.85	−0.22	0.47	1.94	−0.98
Phe	F	−4.22	1.94	1.06	0.54	−0.62
Pro	P	−1.66	0.27	1.84	0.70	2.00
Ser	S	2.39	−1.07	1.15	−1.39	0.67
Thr	T	0.75	−2.18	−1.12	−1.46	−0.40
Trp	W	−4.36	3.94	0.59	3.44	−1.59
Tyr	Y	−2.54	2.44	0.43	0.04	−1.47
Val	V	−2.59	−2.64	−1.54	−0.85	−0.02

## Data Availability

Data will be provided upon request.

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
