# Peer review of "Biodirected Screening and Preparation of Larimichthys crocea Angiotensin-I-Converting Enzyme-Inhibitory Peptides by a Combined In Vitro and In Silico Approach"

_molecules, 2024, doi:10.3390/molecules29051134_

Round 1

Reviewer 1 Report

Comments and Suggestions for Authors

In this paper, Yang et al. screened and characterized peptides with ACE inhibitory function from Larimichthys crocea protein through in vitro and silico methods. By this method, the authors screened 6 peptides had ACE inhibitory function with low IC50 values, indicating that this method is promising for the discovery of active peptides. The manuscript is well-written with very few issues with it. Here are some comments on this study:

1.        Lines 39-40 “Patients and medical professionals alike will prefer naturally derived ACE inhibitory peptides as health self-awareness grows.” It is advisable to explain why the patient prefers the peptide.

2.        Line 75 “LCP”, please define the abbreviation when first used in the introduction.

3.        The figures in the manuscript are of low resolution.

4.        Is it possible to discuss the six novel peptides compared to other functional peptides?

5.        Line 270 “This result is in agreement with the study of [34]”, details of reference 34 should be indicated.

Author Response

molecules-2872951: Biodirected screening and preparation of Larimichthys crocea ACE peptides by a combined in vitro and in silico approach

Response to Reviewer 1 Comments:

In this paper, Yang et al. screened and characterized peptides with ACE inhibitory function from Larimichthys crocea protein through in vitro and silico methods. By this method, the authors screened 6 peptides had ACE inhibitory function with low IC50 values, indicating that this method is promising for the discovery of active peptides. The manuscript is well-written with very few issues with it. Here are some comments on this study:

On behalf of my co-authors, we thank you very much for giving us an opportunity to revise our manuscript (molecules-2872951). We appreciate your summary of the manuscript and encouraging comment. All of the comments are valuable and very helpful for revising and improving our manuscript; they also provide important guiding significance for our research studies. We have discussed all of the comments carefully, have made some corrections, and hope the revised manuscript will meet with your approval. The modifications are shown in red. Specific responses to the reviews are listed below:

Q1:Lines 39-40 “Patients and medical professionals alike will prefer naturally derived ACE inhibitory peptides as health self-awareness grows.” It is advisable to explain why the patient prefers the peptide.

A1:Thanks for the reviewer’s comments. We have added explanations in the appropriate sections (Lines 45-48).

Q2:Line 75 “LCP”, please define the abbreviation when first used in the introduction.

A2:Thanks for the reviewer’s comments. It has been revised accordingly (Line 93).

Q3:The figures in the manuscript are of low resolution.

A3: Thanks for the reviewer’s comments. We speculate that it may be because the document format is so large that when uploaded to the system it gets compressed, resulting in a lower quality figures. We sincerely thank you for your comments, which will help improve the quality of our manuscripts. Meanwhile, we have adjusted the figures in the manuscript accordingly to enhance their quality.

Q4: Is it possible to discuss the six novel peptides compared to other functional peptides?

A4: Thanks for your professional suggestions. Your suggestions have been of great help in improving the quality of manuscript, and we have made some additions (Lines 209-219 and Lines 259-263) based on your valuable suggestions. Meanwhile, we also think it's an excellent suggestion, and we'll continue to research these six peptides in future studies.

Q5: Line 270 “This result is in agreement with the study of [34]”, details of reference 34 should be indicated.

A5: Thanks for the reviewer’s comments. We have added the appropriate supplement (Lines 303-305).

Reviewer 2 Report

Comments and Suggestions for Authors

Totally speaking, this article is regarding the production, and isolation of the angiotensin-I converting enzyme (ACE) inhibitory peptides originated by the technological process of in vitro and in silico hydrolysis of marine fish proteins. The current study's goal was to screen and/or perform prediction of ACE peptides from Larimichthys crocea by bioinformatics coupled with experiment verification. Researchers looked at the molecular docking, stability, degree of hydrolysis, in vitro activity, and inhibition kinetics of bioactive peptides during the real hydrolysis process. Six novel ACE inhibitory peptides were screened. The manuscript is suitable for the Molecules journal and the proposed Special Issue. However, there are some main points that require clarification.

(1) Affiliation section: Please include the matching e-mail addresses of the co-authors listed above, as well as their initials in parenthesis. Please, see the instructions for authors.

(2) Abstract section

There is a lack of a clearly written research goal, the reason for this kind of research, as well as the main objectives of the research. The methodology needs to be rewritten, specifically specifying which analysis was used to perform enzymatic hydrolysis (a type of proteolytic enzyme) of the used protein via an in silico prediction process to determine the bioactivities of the peptides. Obtained experimental results are missing in the whole section. It is necessary to amend and supplement the Abstract section.

(3) Introduction section

Please complete the introduction with specific statistical data on the utilization and production of whole marine protein and bioactive peptides from that protein source, as well as their production, isolation, and purification technological methods. The nutritional composition (especially protein content) of marine fish protein (Larimichthys crocea) must be listed in detail, including the citation of the necessary literature.

Next, isolate the procedures for obtaining peptide fractions with improved bioactivities (i.e., a few illustrations of how the authors enhanced the characteristics of peptides using various endo- and egzo-peptodases).

Next, describe the procedure of in silico analysis a little better (in general, for example, like this: 1) The protein sequence is inserted; 2) it is acted on by enzymes such as plant and gastrointestinal enzymes because they are generally the only ones represented in the bases; 3) a search for the characteristics of the obtained peptides is conducted in order to make the readers who are not familiar with the subject more easily introduced to the results and the methodology used.

(4) Materials and Methods section

Line 289: Delete the term peptide from the subsection heading.

Line 294: What is the meaning of the term material-liquid ratio? Does it refer to the ratio of the mass of the enzyme to the mass of the protein-rich substrate, or something else? Since it is not written clearly enough, I ask you to write the value of the E/S ratio for performing the reaction, using the protein content in crude extracts, not the content of crude extracts for the calculation.

Lines 297-300: The indication of the molecular mass range of the obtained fractions is incorrect. Please correct the test. And please consider if you could really use a 0.5 kDa membrane for ultrafiltration. I am not aware of Vivavlow modules with a pore diameter (cut-off) of less than 1 kDa! Likewise, if you used the cross-flow type of ultrafiltration (and you did as soon as you specified Vivaflow modules), please specify the flow rate and the duration of the flow.

The names of the devices (centrifuge and appropriated rotors and conditions; freeze-dryer, etc.) that were used in the whole experimental work and their manufacturers must be mentioned.

(5) It is advised that the authors recheck the main text during the revision to make this manuscript more readable.

Author Response

molecules-2872951: Biodirected screening and preparation of Larimichthys crocea ACE peptides by a combined in vitro and in silico approach

Response to Reviewer 2 Comments:

Totally speaking, this article is regarding the production, and isolation of the angiotensin-I converting enzyme (ACE) inhibitory peptides originated by the technological process of in vitro and in silico hydrolysis of marine fish proteins. The current study's goal was to screen and/or perform prediction of ACE peptides from Larimichthys crocea by bioinformatics coupled with experiment verification. Researchers looked at the molecular docking, stability, degree of hydrolysis, in vitro activity, and inhibition kinetics of bioactive peptides during the real hydrolysis process. Six novel ACE inhibitory peptides were screened. The manuscript is suitable for the Molecules journal and the proposed Special Issue. However, there are some main points that require clarification.

On behalf of my co-authors, we thank you very much for giving us an opportunity to revise our manuscript (molecules-2872951). We appreciate your positive and constructive comments and suggestions on our manuscript. All of the comments are valuable and very helpful for revising and improving our manuscript; they also provide important guiding significance for our research studies. We have discussed all of the comments carefully, have made some corrections, and hope the revised manuscript will meet with your approval. The modifications are shown in red. Specific responses to the reviews are listed below:

Q1: Affiliation section: Please include the matching e-mail addresses of the co-authors listed above, as well as their initials in parenthesis. Please, see the instructions for authors.

A1: Thanks for the reviewer’s comments. We have made modifications accordingly (Lines 6-7, Line 11, 13 and 14).

Q2:(2) Abstract section

There is a lack of a clearly written research goal, the reason for this kind of research, as well as the main objectives of the research. The methodology needs to be rewritten, specifically specifying which analysis was used to perform enzymatic hydrolysis (a type of proteolytic enzyme) of the used protein via an in silico prediction process to determine the bioactivities of the peptides. Obtained experimental results are missing in the whole section. It is necessary to amend and supplement the Abstract section.

A2: Thanks for the reviewer’s comments. We've rewritten the abstract based on your suggestions(Lines 15-30). 

Abstract: Food-derived angiotensin-I converting enzyme (ACE) inhibitory peptides have gained attention for their potent and safe treatment of hypertensive disorders. However, there are some limitations of conventional methods for preparing ACE inhibitory peptides. In this study, in silico hydrolysis, the quantitative structure-activity relationship (QSAR), LC-MS/MS, inhibition kinetics and molecular docking were used to investigate the stability, hydrolyzability, in vitro activity, and inhibition mechanism of bioactive peptides during actual hydrolysis process. Six novel ACE inhibitory peptides were screened from the Larimichthys crocea protein (LCP) and had low IC50 values (from 0.63 ± 0.09 µM to 10.26 ± 0.21 µM), which were close to the results of the QSAR model. After in vitro gastrointestinal simulated digestion activity of IPYADFK, FYEPFM and NWPWMK was found to remain almost unchanged, whereas LYDHLGK, INEMLDTK and IHFGTTGK were affected by gastrointestinal digestion. Meanwhile, the inhibition kinetics and molecular docking results were consistent that ACE inhibitory peptides of different inhibition forms could effectively bind to the active or noncentral active centers of ACE through hydrogen bonding. Our proposed method has better reproducibility, accuracy and higher directivity than previous methods. This study can provide new approaches for the deep processing, identification and preparation of Larimichthys crocea.

Q3: Introduction section

Please complete the introduction with specific statistical data on the utilization and production of whole marine protein and bioactive peptides from that protein source, as well as their production, isolation, and purification technological methods. The nutritional composition (especially protein content) of marine fish protein (Larimichthys crocea) must be listed in detail, including the citation of the necessary literature.

A3: Thanks for the reviewer’s comments. Your suggestions have been of great help in improving the quality of manuscript, and we have made some additions based on your valuable suggestions (Lines 71-81 and Lines 87-90).

Q4: Materials and Methods section

Line 289: Delete the term peptide from the subsection heading.

A4: Thanks for the reviewer’s comments. We've removed peptide (Line 321).

Q5: Line 294: What is the meaning of the term material-liquid ratio? Does it refer to the ratio of the mass of the enzyme to the mass of the protein-rich substrate, or something else? Since it is not written clearly enough, I ask you to write the value of the E/S ratio for performing the reaction, using the protein content in crude extracts, not the content of crude extracts for the calculation.

A5: Thanks for the reviewer’s comments. It does refer to the ratio of the mass of the enzyme to the mass of the protein-rich substrate. Please forgive us for not expressing it clearly enough, we have re-written this part according to your valuable suggestion (Lines 326-327).

Q6: Lines 297-300: The indication of the molecular mass range of the obtained fractions is incorrect. Please correct the test. And please consider if you could really use a 0.5 kDa membrane for ultrafiltration. I am not aware of Vivavlow modules with a pore diameter (cut-off) of less than 1 kDa! Likewise, if you used the cross-flow type of ultrafiltration (and you did as soon as you specified Vivaflow modules), please specify the flow rate and the duration of the flow.

A6: Thanks for the reviewer’s comments. We do use 0.5 kDa membranes, and our membranes come from Zibo Nanotechnology Co. in China. As you mentioned, we also use a constant flow pump. Meanwhile, we have made additions accordingly in our resubmitted manuscript (Lines 333-338).

Q7: The names of the devices (centrifuge and appropriated rotors and conditions; freeze-dryer, etc.) that were used in the whole experimental work and their manufacturers must be mentioned.

A7: Thanks for the reviewer’s comments. We have added the appropriate supplement. (Lines 328-331 and Lines 338-339).

Q8: It is advised that the authors recheck the main text during the revision to make this manuscript more readable.

Q8: Thanks for the reviewer’s comments. We tried our best improve the manuscript and made some changes to the manuscript. These changes will not influence the content and framework of the paper. The change can be found in the revised manuscript (Lines 97-101, 128, 183, 184, 186, 223, 242, 289, 303, 364, 430-432, 438-439, 441-442, 434-436 and 444-447). We appreciate for Reviewers' warm work earnestly and hope the revised manuscript could be acceptable for you.

Reviewer 3 Report

Comments and Suggestions for Authors

The manuscripit could be accepted after fixing some minor issues. 

- First th figures quality should be enhanced overall.

- For the molcular docking, its better to add the lenth of each force.

- I dont know why you did not add something about the MDs, why you did not perform a dynaim simulation for the best docked conjugates to evauate thier stability. 

Author Response

molecules-2872951: Biodirected screening and preparation of Larimichthys crocea ACE peptides by a combined in vitro and in silico approach

Response to Reviewer 3 Comments:

The manuscripit could be accepted after fixing some minor issues.

We feel great thanks for your professional review work on our manuscript. As you are concerned, there are several problems that need to be addressed. We have discussed all of the comments carefully, have made some corrections, and hope the revised manuscript will meet with your approval. The modifications are shown in red. Specific responses to the reviews are listed below:

Q1: First the figures quality should be enhanced overall.

A1: Thanks for the reviewer’s comments. We speculate that it may be because the document format is so large that when uploaded to the system it gets compressed, resulting in a lower quality figures. We sincerely thank you for your comments, which will help improve the quality of our manuscripts. Meanwhile, we have adjusted the figures in the manuscript accordingly to enhance their quality.

Q2: For the molcular docking, its better to add the lenth of each force.

A2: Thanks for the reviewer’s comments. We also initially tried toadd the lenth of each force, but we found that the interaction between ACE and ACE inhibitory peptides is so much that when all force lengths are shown, some of them will be obscured. Therefore, after analyzing the docking results, we only show the key lengths for the main forces. At the same time, we supplement the description of the bond length in the 3D diagram in the diagram note (Lines 296-297).

Q3: I dont know why you did not add something about the MDs, why you did not perform a dynaim simulation for the best docked conjugates to evauate thier stability.

A3: Thanks for the reviewer’s comments. The aim of our study was to establish an effective method for high-throughput screening of ACE inhibitory peptides, and we have verified the reliability of the method through in silico, in vitro validation and mechanistic studies. As you commented, MDs is mainly to study the stability of peptide-inhibitor complex binding to better investigate the mechanism of peptide inhibition. We will use MD to further study the mechanism and physicochemical properties of these six peptides at a later stage.

Reviewer 4 Report

Comments and Suggestions for Authors

The work presented in this manuscript discribes a potentialy diferent way of predictibe ACE inhibitory peptides arrising from larimichthys crocea. The work is good but some modifications are necessary to make it more acurate:

1) ace peptides should be refered as ACE inhibitory peptides throughout.

2) in objectives try to keep the same tense. Each objective needs to be read as a stand alone phrase 

3) the lables of figures should be more discritive, for example mentioning that the values represent the average of n=3 and the error bars are std dev?!

4) in material and methods the contentration thay was used for the ACE inhibitory test needs to be mentioned (for the one showed in % inhibition, the IC 50 is fine)

5) conclusions must be read by a native english speaker to improove readability.

Author Response

molecules-2872951: Biodirected screening and preparation of Larimichthys crocea ACE inhibitory peptides by a combined in vitro and in silico approach

Response to Reviewer 4 Comments:

The work presented in this manuscript discribes a potentialy diferent way of predictibe ACE inhibitory peptides arrising from larimichthys crocea. The work is good but some modifications are necessary to make it more acurate:

On behalf of my co-authors, we thank you very much for giving us an opportunity to revise our manuscript (molecules-2872951). We appreciate your positive and constructive comments and suggestions on our manuscript. All of the comments are valuable and very helpful for revising and improving our manuscript; they also provide important guiding significance for our research studies. We have discussed all of the comments carefully, have made some corrections, and hope the revised manuscript will meet with your approval. The modifications are shown in red. Specific responses to the reviews are listed below:

Q1:1) ace peptides should be refered as ACE inhibitory peptides throughout.

A1: Thanks for the reviewer’s comments. We have made modifications accordingly (Line 3, 128, 183, 184, 186, 223 , 242 and 364).

Q2: in objectives try to keep the same tense. Each objective needs to be read as a stand alone phrase.

A2: Thanks for the reviewer’s comments. We have made adjustments accordingly (Lines 97-101).

Q3: the lables of figures should be more discritive, for example mentioning that the values represent the average of n=3 and the error bars are std dev?!.

A3: Thanks for the reviewer’s comments. We have made modifications accordingly (Lines 129 and 131).

Q4:in material and methods the contentration thay was used for the ACE inhibitory test needs to be mentioned (for the one showed in % inhibition, the IC50 is fine)

A4: Thanks for the reviewer’s comments. We have added the appropriate supplement (Lines 357-360).

Q5:conclusions must be read by a native english speaker to improove readability.

A5: Thanks for the reviewer’s comments. We apologize for the poor language of our manuscript. We worked on the maruscript for a long time and the repeated addition and removal of sentences and sections obviously led to poor readability. According to your nice suggestions, We invited the Elsevier Language Editing to help polish our manuscripts. These changes will not influence the content and framework of the paper (Lines 430-446). And we also revised other incorrect sentences in the manuscript (Lines 97-101, 128, 183, 184, 186, 223, 242, 289, 303, 364, 430-432, 438-441, 433-435 and 443-446). We appreciate for Reviewers' warm work earnestly and hope the revised manuscript could be acceptable for you.